# Cellular Therapies for Multiple Myeloma: Engineering Hope

**DOI:** 10.3390/cancers16223867

**Published:** 2024-11-19

**Authors:** Sarah Vera-Cruz, Maria Jornet Culubret, Verena Konetzki, Miriam Alb, Sabrina R. Friedel, Michael Hudecek, Hermann Einsele, Sophia Danhof, Lukas Scheller

**Affiliations:** 1Medizinische Klinik und Poliklinik II und Lehrstuhl für Zelluläre Immuntherapie, Medizinische Klinik II, Universitätsklinikum Würzburg, 97080 Würzburg, Germany; 2Fraunhofer-Institut für Zelltherapie und Immunologie (IZI), Außenstelle Zelluläre Immuntherapie, 97080 Würzburg, Germany; 3Mildred Scheel Early Career Center, Universitätsklinikum Würzburg, 97080 Würzburg, Germany; 4Interdisziplinäres Zentrum für Klinische Forschung (IZKF), Universitätsklinikum Würzburg, 97080 Würzburg, Germany

**Keywords:** multiple myeloma, cell therapy, CAR-T, CAR-NK, clinical trial, BCMA, GPRC5D

## Abstract

Cellular immunotherapy represents a rapidly advancing field in cancer treatment, particularly for multiple myeloma (MM). Notably, two approved chimeric antigen receptor (CAR) T cell therapies targeting B cell maturation antigen (BCMA) have demonstrated substantial clinical efficacy, offering renewed hope for patients with this hitherto incurable disease. However, achieving sustained remission across a wider patient population remains challenging. This review examines the landscape of current clinical trials in MM cellular immunotherapy, highlighting key challenges and opportunities. It also explores recent innovations aimed at overcoming existing barriers, including broadening the spectrum of used cell types, optimizing manufacturing processes and targeting novel antigens. Together, these strategies could improve clinical outcomes, paving the way to push immunotherapy to earlier lines of MM treatment.

## 1. Introduction

Multiple myeloma (MM) is a hematological malignancy characterized by the clonal proliferation of abnormal, terminally differentiated plasma cells in the bone marrow (BM) [1]. Symptoms of MM are often non-specific and usually only appear with the onset of end-organ damage such as bone destruction, anemia, increased susceptibility to infections, kidney failure and hypercalcemia [2]. MM typically evolves from pre-malignant conditions such as monoclonal gammopathy of undetermined significance (MGUS) and smoldering MM (SMM) [3]. Genetic alterations drive the progression of these precursor stages to MM [4]. This complex, multistage process obscures and prolongs the diagnostic timeline, often with many years elapsing between the suspected initial genetic event and the time point of diagnosis [5,6].

MM is the second most common hematological malignancy (10–15%), with its incidence rising since 1990; this condition mainly affects the elderly, with a median patient age of 65 to 70 at the time of diagnosis [7,8]. Demographic changes in society and advances in diagnostics and treatment have resulted in increasing disease prevalence and impact [9].

Although many effective treatment methods have appeared in the past 10 years, including monoclonal antibodies (mABs), proteasome inhibitors (PIs) and immunomodulators (IMiDs), MM remains largely incurable. However, the therapeutic landscape has significantly expanded with the introduction of cellular therapies [10]. The FDA approvals of the chimeric antigen receptor T (CAR-T) cell products idecabtagene vicleucel (ide-cel) and ciltacabtagene autoleucel (cilta-cel) have broken new ground for diverse cellular treatment options not only targeting different MM antigens but also benefiting from other cell types, such as natural killer (NK) cells and dendritic cells (DCs) [11,12,13,14].

## 2. Types of Cell-Based Therapies for MM

### 2.1. CAR-T Cells

CAR-T cell therapy has emerged as a promising treatment for MM due to its success against B cell malignancies [15,16]. The innovative approach involves the genetic modification of a patient’s own T cells to express a receptor targeting specific tumor antigens, enabling effective cytolysis of tumor cells in a major histocompatibility complex (MHC)-independent manner. The CAR construct comprises three main building blocks: an extracellular antigen recognition domain often consisting of a single-chain fragment variant (scFv) derived from an antibody, a transmembrane domain and an intracellular domain [17]. The rapid development of this technology in recent years has resulted in five generations of CAR-T cell therapy thus far, with each generation incorporating additional modifications of the intracellular moiety in an attempt to increase the therapeutic potential and decrease adverse effects. To overcome the low expansion and short persistence of first-generation CAR-T cells, which only contained a CD3ζ domain, a costimulatory domain was incorporated in the second generation of CARs. This was further refined in the third generation, which featured two costimulatory domains within a single CAR construct. The fourth generation introduced constitutively or inducibly expressed chemokines, and the fifth generation included additional intracellular domains derived from cytokine receptors [18,19].

### 2.2. CAR-Natural Killer Cells

Although no FDA-approved therapy is yet available, several clinical trials are under way to assess the efficacy of NK cell therapies for MM treatment [20]. As an integral component of the innate immune system, NK cells have been shown to have significant anti-myeloma activity [21]. However, challenges such as the decreased frequency and activity of NK cells in MM patients impair their efficacy, making allogeneic NK cell therapy an attractive approach to restore NK cell functions [22]. Additionally, the development of CAR-NK cells combined the intrinsic tumor-killing capacity of NK cells with enhanced antigen-specific killing ensured by the CAR [23]. Given the absence of T cell receptors (TCRs), graft-versus-host disease (GvHD) is avoided, clearing the way for allogeneic approaches [24]. The short lifespan of NK cells reduces the risk of toxicities; however, along with limited homing capacity and persistence, it also poses a major challenge for clinical translation of NK cell-based therapies [25].

### 2.3. Dendritic Cell Vaccines

A groundbreaking approach in treating MM is the development of DC vaccines, which aim to boost patients’ immune responses and could work synergistically with other MM therapies. DCs are master antigen-presenting cells existent throughout all tissues, capable of capturing, processing and cross-presenting antigens inducing both cellular and humoral immune responses [12]. MM patients often exhibit a quantitative and functional deficiency of DCs, which disrupts antigen presentation and compromises the body’s anti-tumor response [26,27]. Therefore, DC therapies are a promising approach to restore these functions. After DC manufacturing, cells are either loaded with idiotypic proteins or myeloma-associated antigen mRNA or fused with whole MM cells for presentation in an attempt to overcome DC deficiencies and immunosuppression [28,29,30,31,32].

## 3. Cellular Therapies for Treatment of MM

Currently, the approved cellular therapies for treatment of MM are limited to CAR-T cell therapy targeting B cell maturation antigen (BCMA, TNFRSF17). However, several ongoing trials explore new targets and innovative cellular therapies, including NK cells and DCs. Both approved cell products and cell products under development will be discussed in this chapter and are summarized in Table 1. The eligibility criteria for selecting the clinical trials included a systematic search on ClinicalTrials.gov using the keywords CAR-T, CAR-NK and DC vaccines in MM. Trials were prioritized based on published results and/or their significant novelty in the field, as well as follow-up studies from previously published studies.

### 3.1. BCMA CAR-T Cells

BCMA is a type III transmembrane protein that is preferentially expressed in cells of B lineage and is highly abundant on the surface of both malignant and non-malignant plasma cells [66,67]. By binding to its natural ligands, B cell-activating factor (BAFF) and/or proliferation-inducing ligand (APRIL), BCMA induces B cell proliferation and survival as well as maturation and differentiation into plasma cells. BCMA is rarely expressed in other tissues, making it a suitable target for MM [67]. There are currently two CAR-T cell products approved for the treatment of MM, namely, ide-cel and cilta-cel, both targeting BCMA.

### 3.2. Approved BCMA-CAR-T Cell Products

Ide-cel was the first approved CAR-T cell product; it was approved by the FDA and EMA in 2021 for the treatment of MM after three lines of therapy based on the pivotal KarMMa trial [33]. Ide-cel is manufactured from autologous peripheral blood mononuclear cells (PBMCs), transduced with a lentiviral vector containing the BCMA-directed CAR composed of a murine anti-BCMA scFv fused to a CD8α-derived hinge and transmembrane region, a 4-1BB costimulatory domain and a CD3ζ-derived signaling domain [67].

Within the phase II KarMMa trial, ide-cel was evaluated in triple-class-exposed relapsed/refractory MM (RRMM) patients after three previous regimens. A total of 128 patients were treated with ide-cel at target doses of 150 to 450 × 10^6^ total CAR-T cells. Of all treated patients, 73% achieved an overall response (OR), with 33% reaching complete response (CR)/stringent CR (sCR) and 26% achieving minimal residual disease (MRD) negativity. The median progression-free survival (PFS) duration was 8.8 months for all patients, 12.1 months for those with a target dose of 450 × 10^6^ CAR+ T cells and 20.2 months for patients who reached a CR/sCR. In terms of safety, 91% developed neutropenia, 84% cytokine release syndrome (CRS) (5% grade ≥ 3) and 18% immune effector cell-associated neurotoxicity syndrome (ICANS) (3% grade ≥ 3). Thus far, real-world data have shown comparable efficacy and safety for ide-cel in the standard-of-care setting [68]. The subsequent phase III KarMMa-3 trial compared ide-cel to standard regimens in RRMM patients after two to four lines of therapy, including immunomodulatory drugs (IMiDs), proteasome inhibitors (PIs) and daratumumab [36]. In this trial, ide-cel significantly extended PFS (13.3 months for ide-cel vs. 4.4 months for standard of care (SOC)) and improved response rates (OR 71% vs. 42%, CR 39% vs. 5%) compared to standard regimens. Surprisingly, although ide-cel was used in earlier lines in KarMMa-3, the safety profile, ORR and PFS were similar to those found in the KarMMa study. Overall, the KarMMa-3 trial showed extremely poor outcomes for the SOC arm, highlighting the need for novel therapies, and at the same time proved CAR-T (ide-cel) to be a potent one-shot therapy to achieve deep and durable remissions. Based on the results of KarMMa-3, the FDA and EMA extended the approval of ide-cel in April 2024 for treatment of triple-class-exposed RRMM patients after two prior lines of therapy.

Cilta-cel, on the other hand, is produced from purified T cells transduced with a lentiviral vector of a BCMA-directed CAR consisting of two different camelid variable heavy-chain-only domains (VHHs), fused to a CD8α-derived hinge and transmembrane region, a 4-1BB costimulatory domain and a CD3ζ-derived signaling domain. Cilta-cel was first approved in 2022 for the treatment of MM after three lines of therapy based on the results of the pivotal CARTITUDE-1 trial [39,69,70].

The phase Ib/II trial CARTITUDE-1 evaluated cilta-cel in triple-class-exposed RRMM patients after three prior lines of therapy at dose levels ranging from 0.5 to 1 × 10^6^ CAR-T cells/kg body weight. Cilta-cel achieved an impressive OR of 98%, with 83% achieving an sCR and 44% achieving an MRD-negative sCR [70]. The median PFS was 34.9 months, with an estimated overall survival (OS) at 3 years of 62.9% [39]. OR (≥95%) and MRD-negativity (≥80%) rates were high across all subgroups: however, patients with stage III, high-risk cytogenetics, plasmacytomas or high tumor burden (BM plasma cells ≥60%) had a shorter duration of response (DOR) and lower PFS and OS rates [70]. The most common side effects were neutropenia (96%) and CRS (95%; 4% grade ≥ 3). Besides ICANS (17%; 2% grade ≥ 3), other neurotoxicities, including novel movement and neurocognitive treatment-emergent adverse events (MNTs) (5%) of delayed onset, occurred in 12% (9% grade ≥ 3). Cilta-cel was subsequently compared to SOC regimens in lenalidomide-refractory RRMM patients after one to three prior lines of therapy in the phase III CARTITUDE-4 trial [41]. Within this trial, cilta-cel significantly improved PFS (not estimable for cilta-cel vs. 12 months for SOC) and achieved higher OR (85% vs. 67%) and CR (73% vs. 22%) rates than SOC. Strikingly, cilta-cel showed lower rates of CRS and high-grade neurotoxicities, including lower frequencies of MNTs, in CARTITUDE-4 than in CARTITUDE-1. The encouraging results of CARTITUDE-4 led to extension of cilta-cel’s approval by the FDA and EMA in April 2024 for lenalidomide-refractory RRMM patients with progressive disease after at least one prior line of therapy, including a PI and IMiD.

Cilta-cel demonstrated promising long-term results in the first-in-human trial using LCAR B38M (LEGEND-2 (NCT03090659, ChiCTRONH-17012285)) with a 5-year PFS of 21% [38]. However, its higher efficacy comes at the cost of more side effects, including cytopenias and novel late-onset neurotoxicities. In fact, a meta-analysis has shown that cilta-cel was associated with higher non-relapse mortality than ide-cel [71].

Ongoing studies are currently evaluating ide-cel and cilta-cel as first-line therapies in patients with newly diagnosed MM (NDMM). CARTITUDE-5 (NCT04923893) is investigating cilta-cel as a frontline therapy (VRd + cilta-cel vs. VRd + Rd maintenance) in NDMM patients for whom autologous stem cell transplantation (ASCT) is not planned as initial treatment. The EMagine/CARTITUDE-6 trial (NCT05257083) will assess the efficacy and safety of DVRd followed by cilta-cel versus DVRd followed by ASCT in transplant-eligible NDMM patients. KarMMa-4, on the other hand, will study the safety of ide-cel in patients with high-risk (R-ISS stage III) NDMM. The results of these studies are eagerly awaited to clarify the future roles of CAR-T cell therapy and ASCT and whether frontline immunotherapies, especially CAR-T cell therapy, can potentially cure MM.

### 3.3. Other BCMA-CAR-T Cell Products in Clinical Development

Compared to the murine and camelid antigen-binding domains of ide-cel and cilta-cel, several studies are currently investigating fully human or synthetic BCMA-directed CAR-T cells with the aim of increasing persistence by reducing immunogenicity (EVOLVE/NCT03430011, LUMMICAR/NCT03975907, FUMANBA-1/NCT05066646, iMMagine-1/NCT05396885 and NCT03602612). Zevorcabtagene autoleucel (zevor-cel, CT053) and equecabtagene autoleucel (eque-cel, CT103), which both incorporate a fully human BCMA-CAR, showed promising efficacy in the LUMMICAR and FUMANBA trial, with >95% OR; CR rates > 70% (MRD-negativity rates 72–95%) [45,48,72] and mPFS of 25 and 23 months, respectively [44,73]. High-grade CRS occurred in 6.9% within the LUMMICAR study (zevor-cel) and in 1% in the FUMANBA study (eque-cel). In contrast, no high-grade ICANS was observed in either trial. Strikingly, in the initial phase I trial eque-cel (CT103A) had a median CAR-T cell persistence of 419 days [73]. Furthermore, recent data presented at EHA 2024 on the phase I FUMANBA-2 trial investigating eque-cel in high-risk patients with NDMM not eligible for ASCT showed encouraging results with 100% OR, a 94% CR rate, a 100% MRD-negativity rate with 71% achieving sustained MRD negativity for over 12 months and a 12-month PFS rate of 84% [74]. Eque-cel and zevor-cel have been approved in China for the treatment of RRMM after 1–2 prior lines in patients who are refractory to lenalidomide and RRMM after 3 prior lines of therapy, including a PI and IMiD, respectively.

Compared to eque-cel and zevor-cel, anitocabtagene autoleucel (anito-cel, CART-ddBCMA) leverages a completely synthetic d-domain-based antigen-binding domain, which was specifically engineered to reduce immunogenicity and improve CAR cell surface stability [49]. In a phase I trial in heavily pretreated triple-refractory RRMM patients (68% penta-refractory), anito-cel proved to be safe and efficacious (100% OR, 76% CR rates, 56% 24-month PFS rate) [75].

### 3.4. GPRC5D CAR-T Cells

G-protein coupled receptor family C group 5 member D (GPRC5D) is an orphan G-protein coupled receptor that has emerged as a promising target for MM treatment [76]. In contrast to BCMA, GPRC5D is expressed more exclusively on plasma cells than other immune cell subsets, with limited expression in skin and keratinized tissue. GPRC5D CAR-T cell therapy has shown activity both in preclinical [77] and clinical settings, with several ongoing clinical trials.

MCARH109, a GPRC5D-directed CAR-T cell product with a fully human antigen-binding domain, a 4-1BB costimulatory domain and a CD3ζ signaling domain, has been tested in a phase I study in 17 patients with triple-class-exposed RRMM after 3 prior lines of therapy (NCT04555551) [51]. The study enrolled not only many patients with high-risk cytogenetics (76%) but also patients who had undergone prior BCMA-directed therapy (59%), including prior BCMA-directed CAR-T cell therapy (47%). Dose-limiting toxicities (DLT) occurred at a dose of 450 × 10^6^ CAR-T cells, with one patient developing grade 4 CRS and ICANS and two patients developing grade 3 cerebellar disorders. Infections were noted in 18% of patients (12% grade 3). On-target off-tumor toxicities associated with GPRC5D expression in keratinized tissues such as nail changes (65%), rashes (18%) or dysgeusia (12%) were observed, mainly of grade 1. At the maximum tolerated dose of 150 × 10^6^ CAR-T cells, an OR rate of 58% and a CR rate of 25% (50% MRD-negativity rate) was observed. Of note, across all dose levels, OR and CR rates did not differ between patients with and without previous BCMA-directed therapies.

OriCAR-017 utilizes nanobody-based tandem CAR-T cells that target two different epitopes of GPRC5D [78]. The phase I trial (POLARIS), which tested OriCAR-017 in RRMM patients with proven GPRC5D expression and at least three prior regimens (50% with prior BCMA therapy), showed a safe toxicity profile (no DLT or serious adverse events, no high-grade CRS and no ICANS) and encouraging OR rates of 100%, with 80% CR rates and a median PFS of 11.4 months [52]. OriCAR-017 is currently being further investigated in two phase I/II trials (NCT06182696 and NCT06271252).

BMS-986393 (CC-95266) has been investigated in a phase I trial of 70 RRMM patients with at least three prior regimens (36% with prior BCMA therapy) [53]. CRS occurred in 84% (4% grade ≥ 3), with one case of grade 5 CRS. Hemophagocytic lymphohistiocytosis occurred in three patients (all grade 3), ICANS in 11% (3% grade ≥ 3) and neurotoxicities other than ICANS in 4%. These included events termed cerebellar toxicity, paresthesia, gait disturbance and nystagmus. Any-grade infections were noted in 43% of patients (16% grade ≥ 3). On-target off-tumor toxicities related to skin (24%), nails (16%) and dysgeusia (3%) were mild (all grade 1–2). The OR rate was 86% and the CR rate 38% in all patients, and patients refractory to prior BCMA-directed therapies had 85% OR and 46% CR rates. Based on these results, a phase II trial (QUINTESSENTIAL, NCT06297226) is now further evaluating BMS-986393 in RRMM patients.

In summary, the abovementioned trials showed a relatively safe toxicity profile with similar but fewer treatment-related adverse events linked to on-target off-tumor toxicities such as skin reactions and dysgeusia compared to GPRC5D-directed bispecific antibodies [66,76]. However, the observed cerebellar disorders with GPRC5D-directed CAR-T cells need further investigation, since they have not yet been reported with GPRC5D bispecific antibodies. Of note, the incidence of infections appears lower than with BCMA-directed therapies, possibly due to lower GPRC5D expression on immune cells, particularly normal plasma cells and B cells. Given their promising efficacy, especially in patients who had previously been treated with BCMA-targeting therapies, GPRC5D-directed CAR-T cells are emerging as an important addition to the therapeutic spectrum of RRMM. Nevertheless, it should be taken into consideration that GPRC5D has been proposed to exhibit reduced genomic stability and could be more easily lost than BCMA [79]. This highlights the urge for continuing research focusing on addressing the challenges posed by genomic alterations in MM.

### 3.5. SLAMF7/CS1 CAR-T Cells

Signaling lymphocyte activation molecule Family 7 (SLAMF7, CS1, CD319) is a surface antigen that is highly expressed on malignant plasma cells in both NDMM and RRMM patients. Furthermore, recent data suggest that genomic alterations of the BCMA antigen coincide with copy number gains in alternative antigens, including SLAMF7 [79]. Besides its expression on immune cell subsets such as NK, T and B cells, where it mediates activating or inhibitory functions, SLAMF7 is not known to be present in any other normal human tissues, making it a suitable target for CAR-T therapy in MM [80]. CARAMBA-01 was a first-in-human clinical trial studying the safety and feasibility of SLAMF7-directed CAR-T cells manufactured in a virus-free manner by Sleeping Beauty gene transfer in MM patients [81]. Other products using a lentiviral or gamma-retroviral vector to transduce a SLAMF7-directed CAR are in ongoing clinical research (NCT03710421).

### 3.6. CAR-NK Cells

The number of clinical trials investigating CAR-NK cells is exponentially growing. CAR-NK cells can be engineered from different sources including cord blood (CB), peripheral blood (PB), NK cell lines (e.g., NK-92) and induced pluripotent stem cells (iPSCs) [82,83].

An off-the-shelf iPSC-derived BCMA-targeting CAR-NK cell product (FT576) is being evaluated in a phase I clinical trial alone or in combination with daratumumab in patients with RRMM (NCT05182073). The cell product harbors a high-affinity, non-cleavable CD16 Fc receptor (hnCD16), a recombinant fusion of IL-15 and IL-15 receptor alpha to allow autonomous persistence (IL-15RF) and a CD38 knockout to avoid fratricide [84]. Of the nine treated patients, three showed disease regression and two had a confirmed ORR. Furthermore, a patient with five prior lines of therapy achieved a very good partial response (VGPR). Overall, the therapy was safe and well tolerated, without CRS, neurotoxicity or GvHD, emerging as one of the promising novel therapies to eradicate MM [63,85]. Other ongoing clinical trials are studying CB-derived (NCT05008536) and NK-92-derived (NCT03940833) BCMA CAR-NKs.

### 3.7. Dendritic Cell Vaccines

DC vaccines have proven to be a feasible approach, as shown by the FDA-approval of sipuleucel-T for prostate cancer [86]. On this regard, different efforts are being made to boost the immune response in MM.

Survivin is a protein inhibitor of apoptosis overexpressed in almost all cancers while being undetectable in most normal adult tissues. In MM, its expression is related to poor prognosis, disease progression and drug resistance [12,87]. The presence of survivin-specific CD8+ and CD4+ cells in MM supports its immunogenicity and the potential of survivin as a valid target [88,89]. A survivin protein-pulsed DC vaccine administration before and after ASCT was studied in a phase I clinical trial (NCT02851056) involving 13 NDMM patients who did not achieve CR with induction therapy. The results showed that it was safe and immunogenic, with 85% of the patients showing a response, suggesting that this strategy could help improve patient outcomes [64].

In a phase II randomized clinical trial (NCT02728102), MM cells derived from patients were fused with autologous monocyte-derived DCs. The DC product was administered in combination with GM-CSF and lenalidomide compared to lenalidomide and GM-CSF or lenalidomide alone following ASCT. The first results showed that the combination of the DC/MM fusion vaccine and lenalidomide maintenance did not result in a significant increase in CR or VGPR rates after 1 year. However, patients who received the MM/DC fusion therapy demonstrated a significant expansion of tumor-reactive lymphocytes in both PB and BM [65].

## 4. Overcoming Challenges of CAR-T Cell Therapy in Multiple Myeloma

Although cellular therapies have shown promising results in MM therapy, several challenges must be addressed to further improve patient outcomes. These limitations include manufacturing and patient access, associated toxicities, tumor immune resistance supported by an immunosuppressive tumor microenvironment (TME) and T cell dysfunction (Figure 1).

### 4.1. CAR-T Cell Production

CAR-T cell production involves a complex, multistep process that must be conducted under stringent Good Manufacturing Practices (GMP). The primary goal is to achieve high CAR-T cell numbers while ensuring their ability to proliferate and persist within the patient [90,91]. A major challenge is the long manufacturing process, as the time between apheresis and infusion (vein-to-vein time) can be four to five weeks [41,92]. This not only carries the risk of the patients’ condition worsening but also drives up costs and limits its accessibility. Moreover, manufacturing capacities are constrained by the limited availability of production facilities equipped with the necessary infrastructure, apheresis slots and highly trained personnel [93]. Ultimately, long and complex manufacturing processes are a major contributing factor to the socioeconomic concerns arising with this therapy [94].

Several clinical trials are currently exploring novel manufacturing processes to increase patient access and improve cell product phenotype. In a phase I trial (NCT04394650), fully human BCMA-directed CAR-T are produced using the NEX-TTM platform. Due to reduced ex vivo expansion, this method shortens manufacturing to five to six days and yields a cell product with an increased memory phenotype and enhanced cytokine secretion. The interim results after a median follow-up of 4.9 months showed VGPR and CR rates of 60% and 30% [54]. In another phase I trial, BCMA-targeting CAR-T cells were produced in less than two days using the T-ChargeTM platform. Here, CAR-T expansion occurred completely in vivo within the patient. The most common side effects were CRS in 96% (11% grade 3 CRS) and ICANS in 22% of the patients. DLT, mainly neutropenias and elevated lipase levels, were observed in 13% of the patients. Out of 43 patients treated, 98% achieved an OR [47,95]. In a phase I clinical trial for BCMA/CD19 dual-targeting, FasT CAR-T cells are under investigation for the treatment of RRMM. The cells are produced with a shortened concurrent activation–transduction step of 22 to 36 h followed by in vivo expansion. The initial results prove the safety (7% grade 3 CRS, no ICANS) and efficacy (93.1% OR) of this cell product [96].

Another strategy to overcome the challenges of high production costs, diminished T-cell quality in heavily pre-treated patients and socioeconomic barriers is the development of “off-the-shelf” CAR-T. cells In the phase I trial UNIVERSAL (NCT04093596), allogenic BCMA targeting CAR-T cells were generated using a lentiviral vector. To prevent GvHD and increase persistence, knockout of TCR alpha constant (TRAC) and CD52 was performed using the transcription activator-like effector nuclease (TALEN). Of 43 patients, 55.8% achieved a response (34.9% VGPR), and of the 17 patients who received the highest dosage, 25% were in CR/sCR. Only one DLT was observed, and 88% of the patients reported at least one adverse effect (AE), with neutropenia being the most common (69.8%) [55]. Another allogeneic anti-BCMA CAR-T cell product utilizing next-generation CRISPR-Cas12a technology to knock out TRAC and B2M is being investigated in the CAMMOUFLAGE trial (NCT05722418). The CAR cassette is site-specifically integrated into the TRAC locus. Additionally, a B2M-HLA-E fusion transgene is knocked in into the B2M locus to protect the cells from NK cell-mediated toxicity and thereby improve the persistence of the cells [97]. Although promising, extensive genomic modifications carry the risk of unintended genomic alterations, which could potentially lead to malignant transformation of the cell product. Allogeneic CAR-T cell studies using CRISPR-Cas9 and TALEN showed genomic rearrangements in around 5–10% of the cells [98,99,100]. Compared to that, novel strategies such as prime- and base-editing show improved safety profiles by employing a nickase instead of a nuclease and thereby only inducing a single-strand break in the DNA [101].

Currently, all commercially available CAR-T cell products rely on viral gene transfer methods. However, viral vectors are costly and have limited production capacity. A promising alternative, currently being investigated in clinical trials, is non-viral gene transfer using transposon-based systems, such as the Sleeping Beauty (SB) system [81]. SB gene transfer not only offers a favorable safety profile but is also cost-effective, with the potential for scalable production, which could significantly improve patient access. Another strategy to enhance accessibility is being explored in the ARI0002h trial (NCT04309981) [42]. This academic point-of-care anti-BCMA CAR-T cell trial raises the question of whether academia can serve as a cost-efficient, on-site producer of CAR-T cells. It also prompts discussions about the future dynamics between academic institutions and the biopharmaceutical industry in CAR T cell manufacturing.

### 4.2. Immunosuppressive Tumor Microenvironment

One of the main hurdles is the complex TME, which not only restricts cellular infiltration but also hinders anti-tumor immune responses [102]. The TME consists of a heterogeneous population that not only provides strong immunosuppressive signals but also secretes high cytokine levels, contributing to CAR-T cell-associated toxicities. The complex cellular milieu includes several key players: endothelial cells, myeloid-derived suppressor cells (MDSCs), tumor-associated macrophages (TAMs), regulatory T cells (Tregs) and cancer-associated fibroblasts (CAFs), among others [102]. Increasing evidence underscores the role of the immunosuppressive TME in promoting CAR-T cell exhaustion and dysfunction, thereby significantly impairing clinical outcomes [103,104].

To counteract the immunosuppressive effects of the TME, novel approaches and targets are being developed to modulate the TME and thereby boost the efficacy of cellular therapies. One promising approach focuses on targeting CAFs, which support the immunosuppressive nature of the BM niche. CAFs recruit different innate and adaptive immune cells, such as Tregs and MDSCs, and are also assumed to remodel the extracellular matrix and upregulate the secretion of immunosuppressive cytokines [105,106,107]. Sakemura and colleagues made a breakthrough in elucidating the interplay between CAFs and anti-BCMA CAR-T. CAFs were shown to exert an inhibitory effect on CAR-T cell therapy via the secretion of inhibitory cytokines and chemokines. To overcome CAF-mediated resistance, a dual-targeting approach was developed: CAR-T cells recognizing both malignant cells via BCMA and CAFs via FAP and SLAMF7 showed enhanced anti-tumor function compared to conventional anti-BCMA CAR-T cells [106].

Tregs and their downstream effects present another promising target to enhance cellular therapies. These cells exert their immunosuppressive effects through several mechanisms. They secrete anti-inflammatory cytokines such as IL10 and TGFß, induce granzyme-dependent cytolysis and cause metabolic dysfunction by depriving effector cells of IL2 via their high affinity receptor CD25. Additionally, Tregs are capable of modulating DC maturation and function [108]. The monoclonal antibody ipilimumab, which targets the inhibitory receptor CTLA-4 on T cells, has been shown to achieve both Treg depletion and mitigation of effector cell exhaustion. Another approach includes the neutralization of the secreted anti-inflammatory cytokine TGFß to diminish immunosuppressive effects [109]. Furthermore, the employment of CAR-T cells targeting the transmembrane activator and calcium modulating ligand (CAML) interactor (TACI) offer a dual benefit since TACI is not only a myeloma target but is also expressed by Tregs [110]. However, targeting Tregs comes with the risk of autoimmunity if performed systemically, emphasizing the need for tumor-site-specific targeting.

In the ongoing effort to enhance the efficacy of cellular therapies via TME modulation, the development of new small molecule inhibitors is another promising approach. One such example is ORIC-533, a selective inhibitor of the ectoenzyme CD73, which is involved in the generation of the immunosuppressive adenosine from adenosine monophosphate (AMP) [111]. Expanding the range of treatment options opens new possibilities for combinatorial approaches, possibly enhancing the efficacy and specificity of cellular therapies and improving clinical outcomes.

### 4.3. Tumor Heterogeneity

Despite the high response rates of anti-BCMA and anti-GPRC5D CAR-T cell therapies, a significant number of patients relapse. One of the main reasons is antigen escape driven by immunoselection of BCMA- or GPRC5D-negative or mutant clones, which can arise from biallelic deletions at the corresponding gene loci [79,112]. To address the challenge of antigen escape, there is an urgent need to identify new antigens that can be targeted either sequentially or simultaneously. Below, several promising antigens will be discussed.

CD70 is a member of the tumor necrosis factor (TNF) family with limited expression on activated normal immune cells. It is expressed in both solid and hematologic malignancies [113,114] and upregulated in high-risk MM subtypes [115]. A full-length CD27 extracellular domain (ECD)-based CAR against CD70 did not show any signs of fratricide and exhibited robust efficacy with minimal toxicity in vitro against MM, making CD70 a promising candidate target for next-line treatment [115]. A phase I clinical trial is currently ongoing for patients with malignant hematologic diseases, including RRMM (NCT04662294).

The NK group 2D (NKG2D) receptor is an activating receptor that recognizes eight ligands that are upregulated in both solid and hematological disorders, including MM, while generally being absent from healthy tissues. This makes anti-NKG2D CAR-T cells a potential cancer therapy [58,116]. Barber and colleagues demonstrated the efficacy of anti-NKG2D CAR-T cells against MM with protective immunity against tumor re-challenge [117]. The THINK trial (NCT03018405) evaluated anti-NKG2D CAR-T (CYAD-01) for myeloid malignancies without preconditioning or bridging therapy, aiming to reduce stress-induced NKG2D ligand expression on healthy cells. The study showed good tolerability but low activity due to challenges such as fratricide and poor cell persistence [58,116]. To this end, an improved version (CYAD-02) is under clinical investigation for AML and myelodysplastic syndromes (MDS) (NCT04167696) [118]. Additionally, the DEPLETHINK study (NCT03466320) explored a preconditioning regimen to improve CYAD-01 persistence in a similar patient population [119].

Cell adhesion molecules (CAMs) mediate the interaction between MM cells and the BM, thereby promoting the proliferation, survival and extramedullary spread of malignant plasma cells. Moreover, this interaction also contributes to MM drug resistance [120]. CAR-T cells targeting CAMs have shown promising results in preclinical settings [121,122,123]. In this regard, a phase I trial (NCT03778346) based on a fourth generation CAR-T cells targeting BCMA, SLAMF7, the CAMs integrin ß7, CD38 and the selectin CD138 within 10 different dual targeting combinations was investigated, showing encouraging results in terms of safety and efficacy when targeting BCMA. However, data on the targeting of CAMs have not been published to date [59].

Another strategy to prevent antigen escape and thus improve CAR-T cell efficacy in MM is dual targeting. One of the approaches is the generation of tandem CARs, in which two binding domains are incorporated into a single CAR construct within the same cell [124]. On this regard, Sun and colleagues, developed a tandem CAR targeting BCMA and CD24. The latter was shown to be expressed by myeloma cells at relapse after BCMA CAR-T cells treatment. This bispecific CAR showed significantly greater tumor control in vivo, targeting not only bulk MM cells but also residual resistant MM cells. Furthermore, it modulated macrophage-related immune surveillance, inducing a macrophage polarization to the tumor-suppressive M1 phenotype [125].

Several other tandem CARs are currently under development for the treatment of RRMM. Clinical trials are exploring the co-targeting of BCMA and GPRC5D (NCT05509530), CD19 (ChiCTR2000033567) or SLAMF7 (NCT04662099) and have thus far presented promising feasibility and safety profiles [60,61,62].

Another approach consists of ligand-based CARs, which are able to bind to multiple receptors on the target cells. One example is based on the BAFF survival factor that binds to the BAFF receptor, TACI and BCMA. The low probability of malignant B cells to downregulate all BAFF receptors make BAFF-targeting CAR-T cells a promising alternative to prevent antigen escape [57,126]. In fact, a non-viral TcBuster BAFF ligand-based CAR-T cell product showed efficacy in vitro and in vivo against several B cell malignancies and is currently under clinical investigation for the treatment of RRMM (NCT05546723) [57]. Moreover, AUTO2 is a clinical trial targeting BCMA and TACI based on a truncated version of their natural ligand APRIL. In a phase I study, patients showed poor response due to suboptimal CAR design [56,110]. Thus, a trimeric form of APRIL (TriPRIL), which better mimics its natural conformation, is currently in the process of clinical development (NCT03287804) [110].

Additionally, other approaches include co-infusion or sequential infusion of two CAR-T cell pools targeting different antigens and co-expression of two CARs with distinct specificities within the same cell, either through separate vectors or through bicistronic constructs [124].

### 4.4. T Cell Dysfunction

One major reason for the failure of CAR-T cell therapy is T cell dysfunction due to exhaustion, differentiation and reduced persistence [127]. Exhaustion and differentiation are closely intertwined and are characterized by transcriptomic, metabolic and epigenetic alterations in T cells, leading to impaired effector function and reduced proliferation potential [128]. CAR-T cell exhaustion is caused by ligand-dependent and -independent chronic signaling, metabolic competition with tumor cells and inhibitory signals from the TME [129,130].

One strategy to reduce T cell exhaustion due to chronic signaling consists of modifications in the CAR design to fine-tune its signaling. A reduction in CAR affinity through mutations in the scFv, has been shown to enhance CAR safety profile, increase proliferation and reduce exhaustion. However, this also increases the risk of relapse by outgrowth of antigen-low tumor cells [129,130]. Alternatively, the choice of the costimulatory molecules influences the phenotype of the cells. It has been shown that CAR-T cells using 4-1BB as costimulatory signal exhibit slower anti-tumor effects compared to CD28, nevertheless they display increased persistence [131]. CD3ζ is the most commonly used signaling motif. However, CARs containing CD3δ, CD3ε or CD3γ have shown enhancement of effector function in vivo and protection against stimulation induced T cell dysfunction [132]. Alternatively, chronic antigen signaling can also be reduced by a reduction of the tumor burden using bridging therapies [133].

In the TME, CAR-T cells compete with other cells for nutrients. In particular, glucose is important for the effector function and proliferation of CAR-T cells. A way to meet the metabolic needs of CAR-T cells is an artificial overexpression of the glucose transporter GLUT1 in these cells. This leads to an enhanced glucose uptake resulting in increased effector function, enhanced persistence and decreased exhaustion [134]. Another challenge for CAR-T in the TME is the expression of inhibitory molecules on tumor cells such as PD-L1 and CTLA4 [135]. The combination of CAR-T cells and immune checkpoint inhibitors such as pembrolizumab is a strategy to overcome this limitation [136]. Alternatively, the knockout of checkpoint molecules in the CAR-T cells using gene editing tools is investigated [137]. However, these approaches have, thus far, mainly been applied for solid tumors.

To reduce differentiation and enrich for a memory phenotype in the final CAR-T cell product different cytokines and supplements during the manufacturing have been investigated. IL-2 is an important cytokine for T cell proliferation and differentiation and is commonly used for the in vitro expansion of CAR-T cells [90,138]. However, IL-2 induces a more differentiated phenotype [139]. In contrast to IL2, the addition of IL-15, IL-7 and/or IL-21 during the manufacturing induces a stem cell memory phenotype and increases the metabolic fitness [140,141,142]. Furthermore, addition of inosine during the manufacturing can induce stemness and enhance effector function [143].

### 4.5. Toxicities

Despite the major clinical success of CAR-T cell therapy in MM, multiple side effects and toxicities have been reported, including CRS and neurotoxicity/ICANS as well as immune cell-associated hematotoxicity (ICAHT) and infectious complications [144].

CRS is a commonly observed inflammatory syndrome in CAR-T cell therapy characterized by increased chemokine and cytokine levels, and it has been reported for all six approved CAR-T cell products [33,69,145]. Activated CAR-T cells produce cytokines and trigger other immune cells. Macrophages have been identified as the main drivers behind CRS, secreting the key cytokines IL-6 and IL-1β [145]. CRS can lead to potential life-threatening complications such as multi-organ failure, pulmonary edema and respiratory failure [146]. Several risk factors for CRS have been identified, including high CAR-T cell doses, CD28 as a costimulatory domain, high tumor burden and a low CD4/CD8 ratio in the infusion product [146]. Treatment strategies for CRS vary by severity and typically include IL-6 receptor antagonists such as tocilizumab, glucocorticoids and vasopressors [147]. Further treatment options involve IL-1 receptor blockade with anakinra [148]. Alternatively, the tyrosine kinase inhibitor dasatinib has also been reported to be effective for the treatment of CRS by inducing a transient rest state in CAR-T cells [149].

Neurotoxicity is thought to occur due to disruption of the blood–brain barrier by endothelium-activating cytokines and microglia activation; however, the exact underlying mechanism is yet to be elucidated [146,150,151]. ICANS is the most common type of neurotoxicity and involve the central nervous system in a nonfocal manner. The clinical manifestation of ICANS is very diverse and includes tremors, aphasia, confusion, impaired attention and cerebral edema. Treatment of ICANS depends on the severity and includes dexamethasone, corticosteroids and methylprednisolone. Alternative strategies involve IL-1 receptor blockade, IL-6 inhibitors, dasatinib and intrathecal chemotherapy [146,149].

In the CARTITUDE-1 trial with cilta-cel MNTs have been reported for the first time. MNTs are very rare, potentially fatal side effects that occur with a median onset time of 27 days after CAR-T infusion. MNTs have a wide variety of symptoms, including movement disorders such as tremors or parkinsonism, cognitive impairments such as amnesia and bradyphrenia and personality changes such as reduced facial expression [152]. A proposed mechanism of MNTs involves on-target-off-tumor targeting of BCMA in the basal ganglia, supported by reports of CAR-T crossing the blood–brain barrier and infiltrating this region [153]. Additional risk factors comprise high tumor burden at baseline, high CAR-T cell peak expansion and previous ICANS or CRS grade ≥ 2 [152]. MNT cases in the CARTITUDE studies have dropped from 5% in CARTITUDE-1 to 0.6% in CARTITUDE-4 despite a larger patient cohort. This might be due to the treatment in earlier lines as well as improved management strategies that have been implemented, including enhanced bridging therapy to reduce baseline tumor burden [41,152]. MNTs have mainly been reported after cilta-cel treatment; however, to date, two cases after ide-cel treatment have also been described [154,155]. Management strategies for MNTs include, besides early supportive measures, the use of steroids for any-grade ICANS, especially in patients with high tumor burden, and tocilizumab for any-grade ICANS with concurrent CRS [156]. Additionally, conventional chemotherapy such as cyclophosphamide and intrathecal chemotherapy is used in case of excessive CAR-T cell expansion [152]. Further investigations are needed not only to unravel the underlying pathomechanisms of these side effects but also to refine treatment algorithms.

## 5. Conclusions

The approval of the BCMA-targeting CAR-T cell products ide-cel and cilta-cel broke new ground for cellular therapies. Ongoing trials demonstrate promising efficacy and safety profiles, paving the way to move CAR-T cells into earlier lines of therapy and increase patient access. Current research focuses on overcoming the inherent challenges of cellular therapies, including effector cell dysfunction, tumor heterogeneity, immunosuppression and toxicities. Additionally, major efforts are under way to expand the spectrum of the cell types used, benefiting also from NK cells and DCs. While CAR-T cell therapy shows promise for achieving long-lasting remission, CAR-NK cells offer the potential of enhanced safety profiles by minimizing the risk of GvHD. This opens the door for allogeneic approaches, increasing patient accessibility. Furthermore, DC vaccines are being developed to enhance MM therapy by effectively sensitizing T cells to target antigens and boosting the patient’s immune response. Recent advancements and developments hold the promise of reaching the so far unreached: the curability of MM.

## Figures and Tables

**Figure 1 cancers-16-03867-f001:**
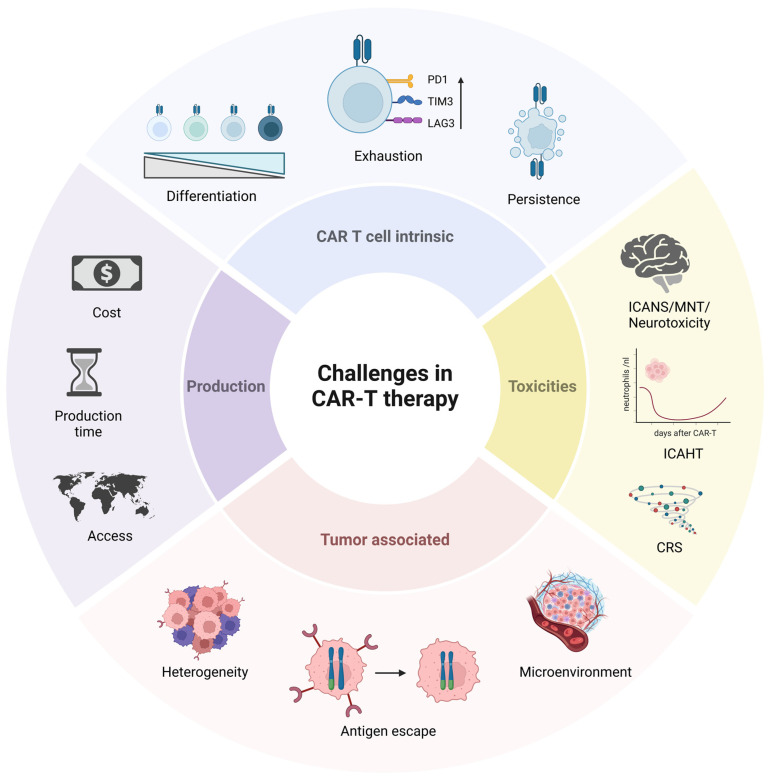
CAR-T cell therapy faces several challenges, such as manufacturing difficulties; CAR-T cell dysfunction and tumor-associated resistance mechanisms, including the immunosuppressive nature of the TME. Additionally, the therapy is associated with various toxicities. Addressing these challenges is crucial for enhancing efficacy and safety of CAR-T treatment. Created with BioRender.com.

**Table 1 cancers-16-03867-t001:** Overview of the current clinical trials from different cellular products for the treatment of multiple myeloma.

Trial Name (Number)	Cell Product	Target	Phase	Product Design	Targeted Population	Treatment	Ref.
**KarMMa (NCT03361748)**	T cell	BCMA	II	Murine- derived	RRMM ≥ 3 prior LOTs (PI, IMiD, anti-CD38 mAb)	Single bb2121 (ide-cel) infusion	[33]
**KarMMa-2 (NCT03601078)**	T cell	BCMA	II	Murine- derived	Cohort 1 (RRMM after ≥3 prior LOTs); cohort 2 (1 prior LOT, PD < 18 mo with (2a) or without (2b) ASCT or with inadequate response post ASCT (2c)); cohort 3 (NDMM with suboptimal response after ASCT)	Cohort 1: ide-cel Cohort 2: ide-cel Cohort 3: ide-cel + lenalidomide maintenance	[34,35]
**KarMMa-3 (NCT03651128)**	T cell	BCMA	III	Murine- derived	RRMM after 2 to 4 prior LOTs (including PI, IMiD, anti-CD38)	Arm A: ide-cel Arm B: DPd or DVd or IRd or Kd or EPd	[36]
**KarMMa-4 (NCT04196491)**	T cell	BCMA	I	Murine- derived	HR (R-ISS Stage III) NDMM after 3 cycles of induction	Single ide-cel infusion	[37]
**KarMMa-7 (NCT04855136)**	T cell	BCMA	I/II	Murine- derived	MM ≥ 3 prior LOTs (PI, IMiD, anti-CD38 mAb) for Arm B and Arm A Cohort 1, after 1–2 prior LOTs for Arm A Cohort 2 (IMiD)	Arm A: ide-cel + CC-220 Arm B: ide-cel + BMS-986405	N/A
**LEGEND-2 (NCT03090659, ChiCTRONH-17012285)**	T cell	BCMA	I/II	Camelid- derived	RRMM ≥ 3 prior LOTs (bortezomib)	Split doses of LCAR-B38M cells	[38]
**CARTITUDE-1 (NCT03548207)**	T cell	BCMA	Ib/II	Camelid- derived	RRMM ≥ 3 prior LOTs (IMiD, PI, anti-CD38 mAb)	Single JNJ-68284528 (cilta-cel) infusion	[39]
**CARTITUDE-2 (NCT04133636)**	T cell	BCMA	II	Camelid- derived	Cohort A (PD after 1–3 prior LOT), cohort B (early relapse after front-line), cohort C (RRMM after PI, IMiD, anti-CD38 and anti-BCMA), cohort D (<CR after front-line ASCT), cohort E (high-risk NDMM with no plans for ASCT), cohort F (standard-risk NDMM with ≥VGPR after initial therapy)	Cohorts A, B, C, F: single cilta-cel infusion Cohort D: cilta-cel + lenalidomide maintenance Cohort E: DVRd induction + cilta-cel + lenalidomide consolidation	[40]
**CARTITUDE-4 (NCT04181827)**	T cell	BCMA	III	Camelid- derived	RRMM after 1 to 3 prior LOTs (lenalidomide refractory)	Arm A: PVd or DPd Arm B: cilta-cel	[41]
**CARTITUDE-5 (NCT04923893)**	T cell	BCMA	III	Camelid- derived	NDMM, not intended to receive ASCT	Arm A: 8c VRd + Rd Arm B: 8c VRd + cilta-cel	N/A
**CARTITUDE-6 (NCT05257083)**	T cell	BCMA	III	Camelid- derived	NDMM, transplant eligible	Arm A: 4c DVRd + ASCT + 2c DVRd + lenalidomide maintenance Arm B: 6c DVRd + cilta-cel + lenalidomide maintenance	N/A
**ARI0002h (NCT04309981)**	T cell	BCMA	I/II	Humanized	RRMM ≥ 2 prior LOTs (IMiD, PI, anti-CD38 mAb)	Single ARI0002h dose	[42]
**EVOLVE (NCT03430011)**	T cell	BCMA	I/II	Fully humanized	Phase I cohort: RRMM ≥ 3 prior LOTs (IMiD, PI, anti-CD38 mAb, ASCT) Phase IIa cohort: RRMM with prior BMCA-directed therapy (anti-BCMA CAR-T cells at least 6 months prior, BCMA-directed engager therapy, BCMA-directed antibody–drug conjugate therapy)	Arm A: JCARH125 (orva-cel) Arm B: JCARH125 (orva-cel) + anakinra	[43]
**LUMMICAR (NCT03975907)**	T cell	BCMA	I/II	Fully human	RRMM ≥ 3 prior LOTs (IMiD, PI, ASCT)	Phase I: single CT053 dose escalation Phase II: single arm (single dose)	[44]
**LUMMICAR-2 (NCT03915184)**	T cell	BCMA	Ib/2	Fully human	RRMM ≥ 3 prior LOTs (IMiD, PI, anti-CD38 mAb)	Phase Ib: single CT053 dose escalation Phase II: single CT053 dose	[45]
**P-BCMA-ALLO1** **(NCT04960579)**	T cell	BCMA	I/Ib	Fully human	RRMM ≥ 3 prior LOTs (IMiD, PI, anti-CD38 mAb)	Part A: P-BCMA-ALLO1 dose escalation +/− Rimiducid Part B: single fixed P-BCMA-ALLO1 dose +/− Rimiducid	[46]
**PHE885 (NCT04318327)**	T cell	BCMA	I	Fully human	Part A cohort: RRMM ≥ 2 prior LOTs (IMiD, PI, anti-CD38 mAb) Part B cohort: NDMM ≥ 4–6 c of VRd, DVRd, DRd	Part A: PHE885 dose escalation Part B: PHE885 dose evaluation	[47]
**FUMANBA-1 (NCT05066646)**	T cell	BCMA	I/II	Fully human	RRMM ≥ 3 prior LOTs (PI, IMiD)	Single CT103A dose	[48]
**Anito-cel (NCT04155749)**	T cell	BCMA	I	D domain	RRMM ≥ 3 prior LOTs (IMiD, PI, anti-CD38 mAb) or “triple-refractory” disease	Single anito-cel dose	[49]
**iMMagine-1 (NCT05396885)**	T cell	BCMA	II	D domain	RRMM ≥ 3 prior LOTs (PI, IMiD, anti-CD38 mAb)	Single arm: anitocabtagene autoleucel	N/A
**NCT03602612**	T cell	BCMA	I	Fully human	NDMM not controlled with standard therapies	Arm A: CAR-T cell dose escalation Arm B: CAR-T cell expansion phase	[50]
**MCARH109 (NCT04555551)**	T cell	GPRC5D	I	Fully human	RRMM ≥ 3 prior LOTs (PI, IMiD, anti-CD38 mAb)	Single MCARH109 dose escalation	[51]
**OriCAR-017 (NCT06182696)**	T cell	GPRC5D	I/II	Fully human	RRMM ≥ 3 prior LOTs (IMiD, PI, anti-CD38 mAb, ASCT)	OriCAR-017 dose escalation followed by dose expansion	[52]
**RIGEL Study (NCT06271252)**	T cell	GPRC5D	I/II	Fully human	Dose escalation phase I: RRMM ≥ 3 prior LOTs Dose expansion phase I and phase II: RRMM (previous BCMA-directed therapies including anti-BCMA bispecific antibody (teclistamab), BCMA-directed antibody conjugate (Blenrep) and BCMA-CAR-T (CARVYKT1TM)	Single OriCAR-017 infusion	N/A
**QUINTESSENTIAL (NCT06297226)**	T cell	GPRC5D	II	Fully human	RRMM ≥ 3 prior LOTs (IMiD, PI, anti-CD38 mAb, anti-BCMA)	Single specified BMS-986393 dose	[53]
**CARAMBA-1 (NCT04499339)**	T cell	SLAMF7	I/II	Humanized	RRMM ≥ 2 prior LOTs (ASCT, IMiD, PI, anti-CD38 mAb)	Single SLAMF7 CAR-T cell dose escalation	N/A
**NCT03710421**	T cell	SLAMF7	I	N/A	RRMM ≥ 3 prior LOTs (IMiD, PI, anti-CD38 mAb, ASCT)	Single SLAMF7 CAR-T dose	N/A
**NCT03958656**	T cell	SLAMF7	I	N/A	RRMM ≥ 3 prior LOTs (IMiD, PI)	Arm A: SLAMF7 CAR-T cell dose escalation Arm B: SLAMF7 CAR-T cell expansion phase	N/A
**CC-98633 (NCT04394650)**	T cell	BCMA	I	Fully human	Arm A and Arm B Cohort A: RRMM ≥ 3 prior LOTs Arm B Cohort B only: RRMM ≥ 1–3 prior LOTs	Arm A: CC-98633 (orva-cel) dose escalation Arm B: CC-98633 expansion phase	[54]
**UNIVERSAL (NCT04093596)**	T cell	BCMA	I	Fully human	RRMM ≥ 3 prior LOTs (IMiD, PI, anti-CD38 mAb)	Arm A: ALLO-715 Arm B: ALLO-715 + nirogacestat	[55]
**CaMMouflage (NCT05722418)**	T cell	BCMA	I	Humanized	RRMM ≥ 3 prior LOTs (IMiD, PI, anti-CD38 mAb)	Arm A: CB-011 dose escalation Arm B: CB-011 expansion phase	N/A
**NCT04662294**	T cell	CD70	I	N/A	Patients with CD70-positive malignant hematologic diseases (AML, NHL, MM)	Single CD70 CAR-T cell dose escalation	N/A
**AUTO2 (NCT03287804)**	T cell	BCMA/TACI	I/II	Murine- derived	RRMM ≥ 3 prior LOTs (IMiD, PI, alkylator)	Phase I: AUTO2 dose escalation phase Phase II: AUTO2 expansion phase selected dose	[56]
**NCT05020444**	T cell	BCMA/TACI	I	Human- derived	RRMM ≥ 3 prior LOTs (IMiD, PI, CD38 mAb) or “triple refractory” disease	Part A: TriPRIL CAR-T cell dose escalation Part B: TriPRIL CAR-T cell dose expansion	N/A
**LMY-920-002 (NCT05546723)**	T cell	BAFF	I	Human	RRMM ≥ 3 prior LOTs (IMiD, PI, anti-CD38 mAb)	LMY-920 single dose escalation	[57]
**THINK (NCT03018405)**	T cell	NKG2D	I/II	Fully human	RRMM	Three cohorts: NKR2 single dose escalation	[58]
**NCT03778346**	T cell	Integrin β7, BCMA, SLAMF7, CD38, CD138	I	N/A	RRMM ≥ 2 prior LOTs	Single dose escalation of different dual target combinations	[59]
**NCT05509530**	T cell	BCMA/GPRC5D	II	Murine- derived	RRMM ≥ 3 prior LOTs (chemotherapy based on bortezomib and/or lenalidomide)	Pre-specified single anti-BCMA/GPRC5D CAR-T cell dose escalation	[60]
**ChiCTR2000033567**	T cell	BCMA/CD19	I/II	Humanized	RRMM ≥ 2 prior LOTs (IMiD, PI)	Single BC19 dose escalation	[61]
**NCT04662099**	T cell	BCMA/SLAMF7	I	Murine- derived	RRMM ≥ 2 prior LOTs (IMiD, PI)	Single anti-BCMA/SLAMF7 CAR-T cell dose escalation	[62]
**FT576 (NCT05182073)**	NK cell	BCMA	I	Derived from scFv human iPSCs	Arm A: RRMM ≥ 3 prior LOTs (IMiD, PI, anti-CD38 mAb) Arm B: RRMM ≥ 2 prior LOTs (IMiD, PI)	Arm A: FT576 Arm B: FT576 + daratumumab	[63]
**NCT05008536**	NK cell	BCMA	I	N/A CB-derived	RRMM ≥ 2 prior LOTs (PI, IMiD)	Single anti-BCMA CAR-NK dose escalation	N/A
**AsclepiusTCG02 (NCT03940833)**	NK cell	BCMA	I/II	N/A NK-92- derived	RRMM	Single anti-BCMA CAR-NK dose escalation	N/A
**NCT02851056**	DC vaccine	Survivin	I	N/A	NDMM	Single DC: AdmS doses before and after ASCT	[64]
**BMT CTN 1401 (NCT02728102)**	DC vaccine	-	II	N/A	NDMM transplant eligible	Arm A: lenalidomide + DC vaccine + GM-CSF Arm B: lenalidomide + GM-CSF Arm C: lenalidomide	[65]

N/A: Not available; LOT: line of therapy; ASCT: autologous stem cell transplantation; IMiD: immunomodulatory drugs; PI: proteasome inhibitor; NDMM: newly diagnosed multiple myeloma; DVRd: daratumumab, bortezomib, Revlimid and dexamethasone; DVd: daratumumab, bortezomib and dexamethasone; PVd: pomalidomide, bortezomib and dexamethasone; DPd: daratumumab, pomalidomide and dexamethasone; Kd: carfilzomib and dexamethasone; IRd: ixazomib, Revlimid and dexamethasone; EPd: elotuzumab, pomalidomide and dexamethasone; rimiducid: safety switch activator.

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
