# Peer review of "Cellular Therapies for Multiple Myeloma: Engineering Hope"

_cancers, 2024, doi:10.3390/cancers16223867_

Round 1
Reviewer 1 Report
Comments and Suggestions for Authors
This review paper is a comprehensive and useful summary of the development status, clinical trial results, new treatment strategies and future directions for various types of immunotherapies for multiple myeloma, including CAR-T cell therapy, CAR-NK cell therapy and dendritic cell therapy.
There are almost no points that need to be corrected, but in Table 1, the letters BCMA, Product, and Target are broken up into different lines, which is a little distracting, so if possible, it would be easier to read if the
Table 1 summarises all the immune cell therapies being used in clinical trials, but I think it would be easier to read if they were separated into chapters, but I'll leave this to the author's judgement.
Author Response
This review paper is a comprehensive and useful summary of the development status, clinical trial results, new treatment strategies and future directions for various types of immunotherapies for multiple myeloma, including CAR-T cell therapy, CAR-NK cell therapy and dendritic cell therapy.
We thank the reviewer for the positive feedback.
Comment 1:
There are almost no points that need to be corrected, but in Table 1, the letters BCMA, Product, and Target are broken up into different lines, which is a little distracting, so if possible, it would be easier to read if the
Table 1 summarises all the immune cell therapies being used in clinical trials, but I think it would be easier to read if they were separated into chapters, but I'll leave this to the author's judgement.
Response 1:
Thank you for the constructive suggestion. We have carefully revised the table by adjusting the font size to avoid any word breaks. We believe that the new formatting enhances clarity and provides a better overview of the different cell therapy strategies, thereby avoiding the need to sub-divide the table into additional chapters.
Reviewer 2 Report
Comments and Suggestions for Authors
The authors present a thorough and complete review on cell therapy in myeloma. The paper is interesting and opportunely focus both on progress and on open questions.
There are only two minor points
Line 187 “Other BCMA-CAR-T under clinical development” In this section, no mention can be found of the characteristic CAR-T related adverse events. Conversely, CRS and ICANS are reported in the equally preliminary studies mentioned in the ensuing section (line 211 and following). The reader could wonder whether these adverse events did not surprisingly occur or were overlooked in the quoted references. Obviously, the authors report what they could find. Nevertheless, a short comment is desirable.
Line 551 “ICANS management depends on the grading and involves dexamethasone, corticosteroids and methylprednisolone.” The sentence is not at the level of the paper
Author Response
The authors present a thorough and complete review on cell therapy in myeloma. The paper is interesting and opportunely focus both on progress and on open questions. There are only two minor points.
Thank you for the positive remarks and helpful suggestions. We have carefully addressed each comment below. All the changes in the text have been highlighted in yellow in the manuscript.
Comment 1:
Line 187 “Other BCMA-CAR-T under clinical development” In this section, no mention can be found of the characteristic CAR-T related adverse events. Conversely, CRS and ICANS are reported in the equally preliminary studies mentioned in the ensuing section (line 211 and following). The reader could wonder whether these adverse events did not surprisingly occur or were overlooked in the quoted references. Obviously, the authors report what they could find. Nevertheless, a short comment is desirable.
Response 1:
We thank the reviewer for pointing this out. We agree with this comment and have added the following part in lines 213-215 to address this issue:
“High-grade CRS occurred in 6.9% within the LUMMICAR study (zevor-cel) and in 1% in the FUMANBA study (eque-cel), whereas no high-grade ICANS was observed”
Comment 2:
Line 551 “ICANS management depends on the grading and involves dexamethasone, corticosteroids and methylprednisolone.” The sentence is not at the level of the paper
Response 2:
Thank you for the comment. We have rephrased the sentence now located in lines 573 and 574 to enhance the stylistic quality:
“Treatment of ICANS depends on the severity and includes dexamethasone, corticosteroids and methylprednisolone.”
Reviewer 3 Report
Comments and Suggestions for Authors
I carefully reviewed the manuscript by Sarah Vera-Cruz et al., titled "Cellular therapies for multiple myeloma: engineering hope". The paper is well-written, in clear and accessible language, and the topic is highly relevant due to the complexities of treatment and relapse risks associated with multiple myeloma. Overall, the paper is executed at a high scientific and technical level.
Major Comments:
-
Lack of a "Materials and Methods" section makes it difficult to assess the comprehensiveness of the paper in covering all cellular therapy approaches for multiple myeloma. It would be helpful to add selection criteria for the reviewed articles so that the reader can understand the basis on which the discussed methods were chosen.
-
The "Discussion" section requires more in-depth analysis. A critical review of each method presented, evaluating its advantages and disadvantages, would be beneficial, along with a discussion on the potential for therapy implementation in terms of patient accessibility and any potential barriers.
Minor Comments:
-
The reference list and in-text citations should be formatted according to the journal's requirements.
-
Table 1 should avoid breaking single letters in column headers.
-
Superscript "10⁶" is missing from the numbers in lines 126, 130, 153, 224, and 228.
-
In the "CAR-T production" section (as well as other places), there is an extra space before the "%" symbol.
I believe that with the correction of these deficiencies, the article can be published in Cancers.
Author Response
I carefully reviewed the manuscript by Sarah Vera-Cruz et al., titled "Cellular therapies for multiple myeloma: engineering hope". The paper is well-written, in clear and accessible language, and the topic is highly relevant due to the complexities of treatment and relapse risks associated with multiple myeloma. Overall, the paper is executed at a high scientific and technical level.
Thank you for the careful review and positive remarks. We have addressed each single comment and highlighted changes in the re-uploaded manuscript in yellow.
Comment 1:
Lack of a "Materials and Methods" section makes it difficult to assess the comprehensiveness of the paper in covering all cellular therapy approaches for multiple myeloma. It would be helpful to add selection criteria for the reviewed articles so that the reader can understand the basis on which the discussed methods were chosen.
Response 1:
Thank you for the constructive comment. To elucidate the criteria used for selecting eligible clinical trials we have added a short part explaining how the search was conducted in lines 113 to 117:
“The eligibility criteria for selecting the clinical trials included a systematic search on ClinicalTrials.gov using the keywords CAR-T, CAR-NK and DC vaccines in MM. Trials were prioritized based on published results and/or their significant novelty in the field, as well as follow-up studies from previously published studies.”
Comment 2:
The "Discussion" section requires more in-depth analysis. A critical review of each method presented, evaluating its advantages and disadvantages, would be beneficial, along with a discussion on the potential for therapy implementation in terms of patient accessibility and any potential barriers.
Response 2:
We thank the reviewer for the valuable suggestion. Although this review article does not include a separate discussion section, we critically discuss different immunotherapy strategies, highlighting their advantages and disadvantages in the section “Types of cell based therapies in MM”. Throughout the manuscript, we carefully analyze several challenges, including patient accessibility (refer to “CAR-T production”) and other factors such as the “Immunosuppressive tumor microenvironment”, “Tumor heterogeneity”, “T cell dysfunction” and “Toxicities”. Additionally, we explore diverse strategies to address these challenges within each chapter. However, we do agree that a concluding overview highlighting the different immunotherapies including their unique advantages would be beneficial. Therefore, we have added the following part to our “Conclusion” in lines 606 to 611:
“While CAR-T therapy shows promise for achieving long time remission, CAR-NK cells offer the potential of enhanced safety profiles by minimizing the risk of GvHD. This paves the way for allogeneic approaches, increasing patient accessibility. Furthermore, DC vaccines are being developed to enhance MM therapy by effectively sensitizing T cells to target antigens and boosting the patient’s immune response.”
Comment 3:
The reference list and in-text citations should be formatted according to the journal's requirements.
Response 3:
Thank you for pointing this out. We have re-formatted the reference list to fulfill the journal’s requirements.
Comment 4:
Table 1 should avoid breaking single letters in column headers.
Response 4:
We thank the reviewer for highlighting this mistake. We have carefully revised the table by adjusting the font size to avoid any word breaks.
Comment 5:
Superscript "10⁶" is missing from the numbers in lines 126, 130, 153, 224, and 228.
Response 5:
Thank you for bringing this to our attention. We have included the superscript.
Comment 6:
In the "CAR-T production" section (as well as other places), there is an extra space before the "%" symbol.
Response 6:
Thank you for pointing this out. We have corrected this.
Reviewer 4 Report
Comments and Suggestions for Authors
Sarah Vera-Cruz’s text is very well written and describes in depth the possible cell therapies for patients with RRMM (CAR-T at different targets, CAR NK, DC-vaccination).
Table 1 helps the reader to understand the results of the various treatment/clinical trials.
The authors also report on the current limitations of CAR technology and possible strategies to overcome the problems of cell product setup and persistence in the host.
The text could benefit from a small introductory section describing possible commonly available therapeutic weapons. This forefounds the rationale for the major effort in finding new therapeutic approaches.
The English language is well written.
Author Response
Sarah Vera-Cruz’s text is very well written and describes in depth the possible cell therapies for patients with RRMM (CAR-T at different targets, CAR NK, DC-vaccination). Table 1 helps the reader to understand the results of the various treatment/clinical trials. The authors also report on the current limitations of CAR technology and possible strategies to overcome the problems of cell product setup and persistence in the host.
Thank you for the positive feedback. We have highlighted the changes to address the comment in the re-submitted manuscript.
Comment 1:
The text could benefit from a small introductory section describing possible commonly available therapeutic weapons. This forefounds the rationale for the major effort in finding new therapeutic approaches. The English language is well written.
Response 1:
We thank the reviewer for the constructive remark. To provide a brief introduction to commonly available therapeutic approaches, we have added the following part, highlighted in lines 57 to 59:
“Although many effective treatment methods have appeared in the past 10 years including monoclonal antibodies (mABs), proteasome inhibitors (PIs), and immunomodulators (IMiDs),…”
Round 2
Reviewer 3 Report
Comments and Suggestions for Authors
I am pleased to see that the authors have thoroughly addressed all my comments and questions. The manuscript has been improved in clarity and depth, and all suggested revisions have been incorporated appropriately. I am confident that the current version of the paper meets the standards for publication and therefore recommend it for publication in Cancers in its present form